# Sequences flanking the core-binding site modulate glucocorticoid receptor structure and activity

Stefanie Schöne[1], Marcel Jurk[1], Mahdi Bagherpoor Helabad[2], Iris Dror[3,†], Isabelle Lebars[4], Bruno Kieffer[4], Petra Imhof[2], Remo Rohs[3], Martin Vingron[1], Morgane Thomas-Chollier[5] & Sebastiaan H. Meijsing[1]

The glucocorticoid receptor (GR) binds as a homodimer to genomic response elements, which have particular sequence and shape characteristics. Here we show that the nucleotides directly flanking the core-binding site, differ depending on the strength of GR-dependent activation of nearby genes. Our study indicates that these flanking nucleotides change the three-dimensional structure of the DNA-binding site, the DNA-binding domain of GR and the quaternary structure of the dimeric complex. Functional studies in a defined genomic context show that sequence-induced changes in GR activity cannot be explained by differences in GR occupancy. Rather, mutating the dimerization interface mitigates DNA-induced changes in both activity and structure, arguing for a role of DNA-induced structural changes in modulating GR activity. Together, our study shows that DNA sequence identity of genomic binding sites modulates GR activity downstream of binding, which may play a role in achieving regulatory specificity towards individual target genes.

[1] Max Planck Institute for Molecular Genetics, Department of Computational Molecular Biology, Ihnestrasse 63-73, Berlin 14195, Germany. [2] Institute of Theoretical Physics, Free University Berlin, 14195 Berlin, Germany. [3] Molecular and Computational Biology Program, Department of Biological Sciences, University of Southern California, Los Angeles, California 90089, USA. [4] Institut de Génétique et de Biologie Moléculaire et Cellulaire (IGBMC), Département de Biologie Structurale, Centre National de la Recherche Scientifique (CNRS) UMR 7104/Institute National de la Santé et de la Recherche Médicale (INSERM) U964/Université de Strasbourg, 1 rue Laurent Fries, BP 10142, 67404 Illkirch Cedex, France. [5] Institut de Biologie de l'Ecole Normale Supérieure, Institut National de la Santé et de la Recherche Médicale, U1024, Centre National de la Recherche Scientifique, Unité Mixte de Recherche 8197, F-75005 Paris, France. † Present address: Department of Biological Chemistry, University of California, Los Angeles, California 90095, USA. Correspondence and requests for materials should be addressed to M.T.-C. (email: mthomas@biologie.ens.fr) or to S.H.M. (email: meijsing@molgen.mpg.de).

Cells can exploit a variety of strategies to ensure that genes are expressed at a specific and well-defined level, including the tight control of the production process of transcripts. The transcription of genes is controlled by the coordinated action of transcriptional factors (TFs), which bind to cis-regulatory elements to integrate a combination of inputs to specify where and when a gene is expressed and how much gene product is synthesized[1]. Signals influencing the level of transcriptional output include the sequence composition of cis-regulatory elements that can, for example, direct the assembly of distinct regulatory complexes (reviewed in refs 2,3). Other mechanisms that influence the transcriptional output of individual genes include the distance of regulatory elements to the transcriptional start site (TSS) of genes[4], the chromatin context in which regulatory elements are embedded[5], DNA methylation[6,7] and post-translational modifications of proteins[1].

For the glucocorticoid receptor (GR), a member of the steroid hormone receptor family, the sequence of its DNA-binding site is known to modulate the receptor's activity. Some studies suggests that depending on the sequence of the GR-binding sequence (GBS), the direction of regulation might be influenced, that is, whether GR will activate or repress transcription[8–11]. Furthermore, the magnitude of transcriptional activation by GR depends on the exact sequence composition of the GBS, which consists of inverted repeats of two half-sites of 6 base pairs (bp) separated by a 3-bp spacer[11]. Affinity for specific GBSs can explain some, but not all, of the modulation of GR activity by the sequence composition of the GBSs[12]. GR activity can also be modulated by DNA shape, which can serve as an allosteric ligand that fine-tunes the structure and activity of GR without apparent changes in DNA binding affinity[13]. GR can 'read' the shape of DNA through non-specific DNA contacts with the phosphate backbone in the spacer region and at other positions within each half-site[11,13]. In addition, GR contacts the minor groove just outside the core 15-bp GBS[11]. How the DNA-induced structural changes in the associated protein result in different transcriptional outputs is largely unknown, but requires an intact dimerization interface and may involve sequence-specific cooperation with GR cofactors[11,13].

Here we further investigated this question and uncovered that the 2 bp flanking the GBS, which are involved in modifying the shape of the DNA target, influence transcriptional output levels. We first studied if GBS variants can modulate GR activity in a chromosomal context and found that GBS variants can indeed modulate GR activity when integrated at a defined genomic locus. Interestingly, this modulation appears to occur downstream of GR binding as the differences in transcriptional responses cannot be explained by differences in occupancy levels based on chromatin immunoprecipitation (ChIP) experiments. Further-more, we analysed genome-wide data on GR binding and gene regulation and identified differences in the sequence composition between GBSs associated with genes with strong and those with weak transcriptional responses to GR activation. Using a combination of experiments with atomic resolution and functional studies, we found that the base pairs directly flanking the core 15-bp GBS modulate GR activity and induce structural changes in both DNA and the associated DNA-binding domain of GR. Together, our studies suggest that modulation of GR activity and structure by GBS variation at positions directly adjacent to the core recognition sequence plays a role in fine-tuning the expression of endogenous target genes.

## Results

**Genomic GR-binding site sequence affects GR activity.** Previous studies relied on transiently transfected reporters to show that GBS composition can modulate GR activity[11,13]. To determine if GBS variants can also influence GR activity in a chromosomal context, we used zinc finger nucleases (ZFN) to generate isogenic cell lines with integrated GBS reporters[14]. The GBS reporters consist of a GBS variant upstream of a minimal promoter driving expression of a luciferase reporter gene (Fig. 1a). Single-cell-derived clonal cell lines with integrated reporters were isolated by flow-activated cell sorting (FACS) and genotyped for correct integration at the *AAVS1* locus (Supplementary Fig. 1A). Consistent with our expectation, no induction by dexamethasone, a synthetic glucocorticoid hormone, was observed for the reporter lacking a GBS (Fig. 1b). For reporters with a single GBS, transcriptional activation was observed with sequence-specific activities ranging from ∼17-fold for the Cgt, to ∼9-fold for the GILZ and ∼2-fold for the SGK2 GBS (Fig. 1b). Notably, activation of the endogenous GR target

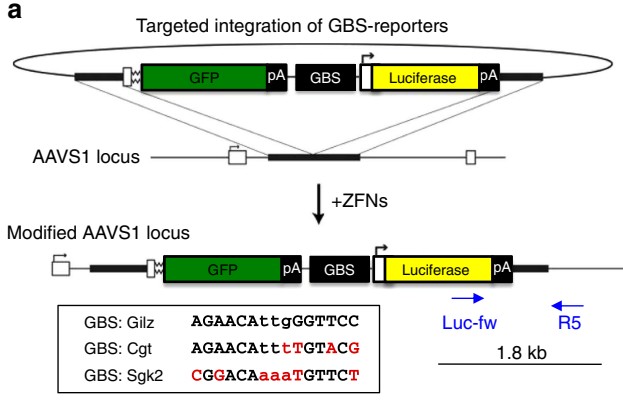

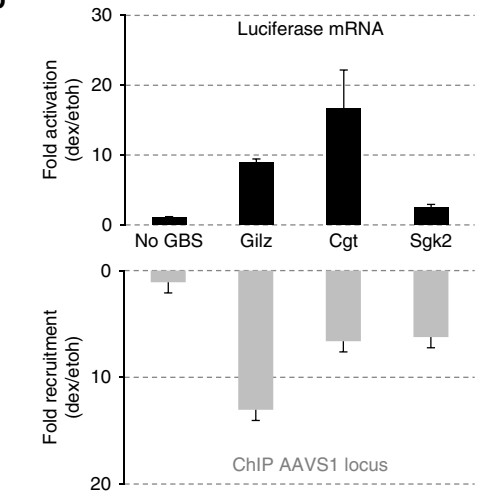

**Figure 1 | GBS activity and binding in a genomic context.** (**a**) Cartoon depicting donor design, GBS sequence and the genotype at the *AAVS1* locus after integration of the GBS-reporters. Nucleotides that diverge from the Gilz sequence are highlighted in red for the Cgt and Sgk2 GBSs, respectively. (**b**) Top: transcriptional activation of the integrated luciferase reporters by GBS variants. Clonal lines with integrated reporters as indicated were treated for 8 h with 1 μM dexamethasone (dex) or 0.1% ethanol as vehicle control. Fold induction of the luciferase reporter gene (dex/etoh) was determined by qPCR. Averages ± s.e.m. are shown (*n* = 3). Bottom: GR binding to GBS reporter variants was quantified by chromatin immunoprecipitation followed by qPCR. Average fold enrichment per reporter variant on dex treatment (1 μM dex, 1.5 h), relative to ethanol vehicle control ± s.e.m. is shown for at least three clonal lines with reporter integration at the desired locus.

gene *TSC22D3* was comparable for all clonal lines (Supplementary Fig. 1B), arguing that the GBS-specific activities are not a simple consequence of clonal variation in GR activity.

To assess if the GBS-specific transcriptional activities could be explained by differences in GR occupancy, we compared GR recruitment to the GBS variants by ChIP. For all clonal lines, a similar level of hormone-dependent GR recruitment was observed for the endogenous *FKBP5* locus, indicating that the ChIP efficiency was comparable between our clonal lines (Supplementary Fig. 1C). As expected, the integrated reporter lacking a GBS showed no GR binding, whereas GR was recruited in the presence of a GBS (Fig. 1b). However, no clear correlation between the level of transcriptional activity and GR recruitment was observed. For instance, the GILZ GBS, which showed an intermediate transcriptional activity, showed the highest occupancy whereas recruitment was comparable for the GBSs with the highest (Cgt) and lowest (Sgk2) activities (Fig. 1b).

Together these data show that GBS nucleotide variation can modulate GR activity in a chromosomal context. Furthermore, this modulation appears to occur downstream of recruitment, consistent with the idea that DNA can change the structure and activity of GR.

**Genome-wide computational analysis of GBS variants**. The experiments with integrated GBS reporters showed that GBS variants can modulate the activity of GR towards target genes in a chromosomal context. To assess whether GBS variants may indeed play a role in fine-tuning the activity of GR towards individual endogenous target genes, we analysed genomic data to see if the level of GR activity correlates with the presence of specific GBS variants near genes. Therefore, we first grouped genes regulated by GR in U2OS cells[15], a human osteosarcoma cell line, into strong responders (top 20% with greatest fold induction on dexamethasone treatment, 290 genes) and a control group of weak responders (genes with significant changes in expression, log2-fold change < 0.72, 688 genes) (Fig. 2a). Next, we associated GR-bound regions, based on ChIP-seq data[15], with a regulated gene when a ChIP-seq peak was located within a window of 40 kb centred on the TSS of that gene (Fig. 2a). The strong GR-responsive genes were associated with 543 peaks. To compare our findings, a control group with similar peak number was generated consisting of 532 peaks that were associated with weak GR responsive genes. For each group of peaks, we conducted a *de novo* motif search with RSAT peak motifs[16]. For both groups, we identified the GR motif (Fig. 2a) and motifs of AP1 and SP1, which are known cofactors of GR[17,18]. The core GR motif was similar for both groups (Fig. 2a) and closely matches the GR consensus sequence[15]. However, we observed subtle differences in preferred nucleotides at individual positions. For instance, the spacer for GBSs associated with weak responders preferentially contains a G or C at position $-1$, whereas no such preference is observed for GBSs associated with strong responders. This is consistent with previous studies showing that the sequence of the spacer can modulate GR activity[11,13]. Furthermore, we found that the nucleotide flanking each half site (position $-8$ and $+8$) exhibited high information content in the strong responders data set, with sequence preferences that were different for peaks associated with strong and weak responder genes (Fig. 2a). For GBSs associated with strong GR responsive genes, the flanking nucleotide was preferentially an A or T, whereas for GBSs associated with weak GR responsive genes the flanking nucleotide was preferentially a G or C. Because the motifs discovered by the *de novo* motif search are not necessarily present at different frequencies in the two groups, we quantitatively compared the

occurrences of motif matches flanked by A/T and G/C nucleotides 5′ and 3′ of the core motif, which are associated with 'strong' and 'weak' peaks, respectively. Consistent with the outcome of the *de novo* motif search, this analysis showed more motif matches for the A/T flanked motif for strong-responder-associated peaks compared with weak-responder-associated peaks, whereas the opposite was found when we scanned with the G/C flanked motif (Supplementary Fig. 2). Together, this suggests that GBS variants may indeed play a role in modulating GR activity towards endogenous target genes, and hint at a possible role in this process for the base pairs directly flanking the half-sites.

**GBS flanking nucleotides modulate GR activity**. To test the role of base pairs flanking the half-site (position $-8$ and $+8$) in modulating GR activity, we generated reporters where we flanked each of five GBS variants (Cgt, FKBP5-1, FKBP5-2, Pal and Sgk) by either A/T or by G/C bp (Fig. 2b). These reporters displayed comparable basal activities, whereas the level of induction on dexamethasone treatment varied between the sequence variants (Fig. 2b). Consistent with the observations for endogenous GR target genes the A/T flanked GBSs showed higher reporter gene activity than the G/C flanked GBSs for four out of five tested GBS variants, whereas little to no effect of changing the flanks was observed for the Pal sequence (Fig. 2b). For example, the activity of A/T flanked Cgt was twice that of the G/C flanked version of this GBS (Fig. 2b). Together, these experiments indicated that the proximal flanking nucleotides can indeed modulate GR activity, and from now on we use the term 'flank effect' to refer to the dependency of GR target gene expression on flanking nucleotides of the GBS core motif. Notably, the Sgk and Cgt GBSs showed the greatest flank effect whereas the effect for the Pal and FKBP5-1 GBSs was small. When comparing the sequences of these GBS variants, we observed that the second half-site (position 2–7) forms an 'imperfect' palindromic sequence (not matching TGTTCT) for the GBSs with the greatest flank effect (Cgt and Sgk) whereas this sequence is palindromic for Pal and FKBP5-1. To test whether the 'imperfect' half-site of Cgt and Sgk is responsible for the flank effect, we generated new luciferase reporter constructs with mixed flanking nucleotides 5′ and 3′ of the core motif (A/C and G/T) (Fig. 2c). These experiments showed that the imperfect half-site is indeed mainly responsible for the flank effect, with on average a 98% increase in activity when we change the flank of the imperfect site, whereas this increase was a more modest 18% when we changed the flank of the 'perfect' half-site.

We focused on the Cgt and Sgk GBS in further experiments as they showed the strongest influence of the flanking nucleotides. To study the role of flanking nucleotides in the chromosomal context, we stably integrated a Sgk-GBS luciferase reporter in U2OS cells at the *AAVS1* locus to simulate an endogenous gene environment. Matching what we observed with the transiently transfected reporters, we again found that integrated A/T flanked Sgk showed a ∼1.5 times greater reporter activity than the G/C flanked GBS (Fig. 3a). At this point, we wondered how the proximal flanks influence GR activity. To determine whether the flank effect might be caused by a change in the intrinsic affinity of the DNA-binding domain (DBD) for GBSs, we conducted electrophoretic mobility shift assays (EMSAs). However, arguing against a role for changes in the intrinsic affinity, we found similar Kd values for both A/T and G/C flanked Cgt and Sgk GBSs (Fig. 3b). In a second approach, we also studied GR binding *in vivo* to A/T and G/C flanked Sgk versions of the stably integrated reporter constructs from the previous experiment by ChIP (Fig. 3a). Remarkably, the GR occupancy of G/C flanked

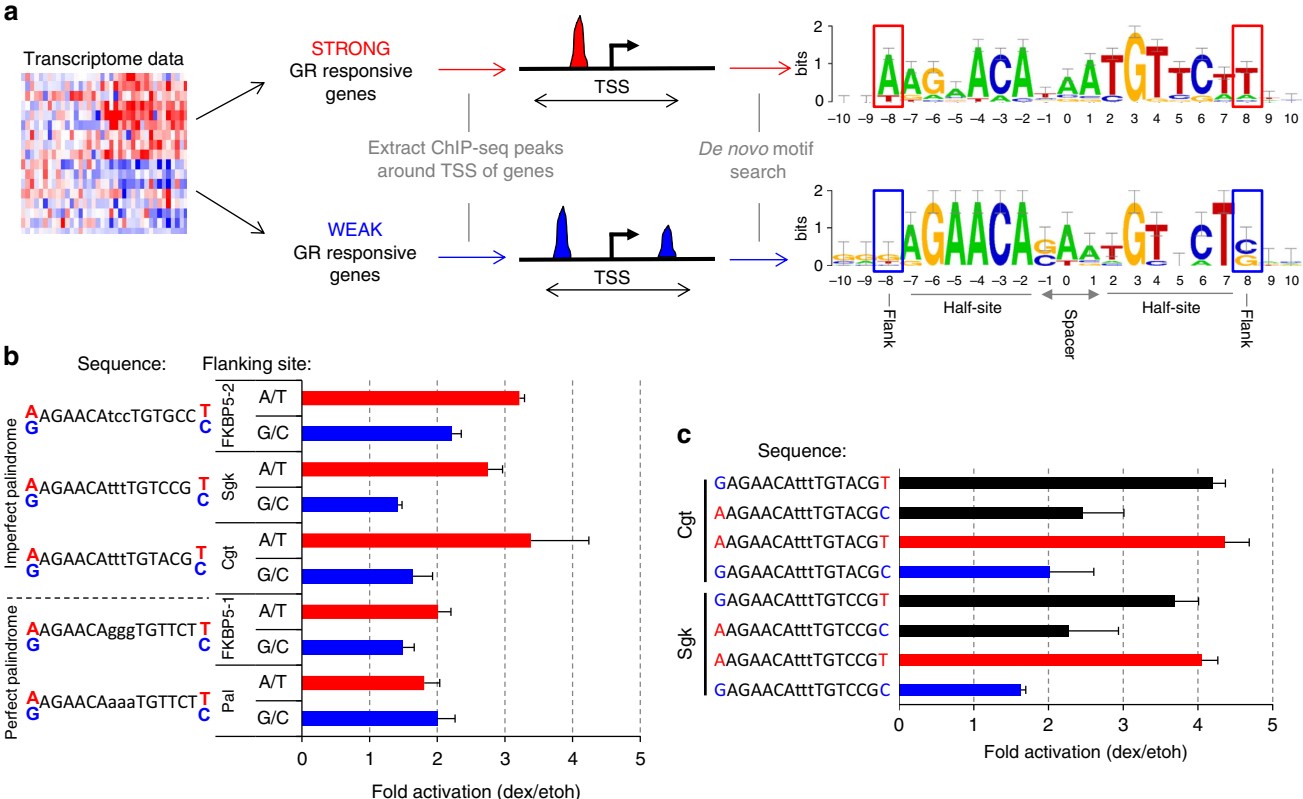

**Figure 2 | Identification and characterization of high-activity GBS variants. (a)** Overview of the workflow to identify candidate high-activity GBS variants. Genes were grouped into strong (top 20% highest fold induction) and weak (log2 fold change < 0.72) transcriptional responders to dexamethasone treatment. Next, ChIP-seq peaks in a 40 kb window centred on the TSS of responder genes were extracted for each group and subjected to *de novo* motif searches resulting in the depicted motifs. The flank positions ($-8$ and $+8$) are highlighted by red (A/T) or blue (G/C) rectangles. **(b)** Transiently transfected luciferase reporter induction of GBS sequences flanked by either A/T or G/C nucleotides. Average fold induction upon 1 µM dexamethasone (dex) treatment relative to ethanol (etoh) vehicle ± s.e.m. ($n \geq 3$) is shown. **(c)** Comparison of transcriptional induction of transiently transfected Cgt and Sgk GBS variants with G/T and A/C 'mixed flanking sites' compared with A/T and G/C flanks. Average fold induction on 1 µM dexamethasone (dex) treatment relative to ethanol (etoh) vehicle ± s.e.m. ($n \geq 3$) is shown.

Sgk was twice that of the A/T flanked Sgk (Fig. 3c), despite the fact that A/T flanked Sgk leads to higher gene activation. Similarly, GR binding was essentially the same when comparing the peak height of all endogenous GR ChIP-seq peaks containing an A/T flanked GBS with those flanked by G/C (Supplementary Fig. 3), showing that peak height and flanking site sequence are independent. Together, we therefore conclude that the flank effect appears not to be a consequence of changes in DNA-binding affinity.

**Flanking nucleotides modulate DNA shape**. Previous studies have shown that the sequence of the spacer influences DNA shape and GR activity[13]. To test whether the local structure of the DNA-binding site is affected by the flanking nucleotides of the GBS, we compared DNA shape features between G/C (75 GBSs) and A/T (83 GBSs) flanked GBSs from peaks associated with weakly and strongly upregulated genes, respectively. The DNA shape features were predicted using a high-throughput method that has been extensively validated based on experimental data[19]. This analysis showed a slight difference in minor groove width between GBSs flanked by G/C and A/T at positions $-8$ and $+8$ (proximal flanks) (Fig. 4a). More strikingly, at positions $-7$, $+7$, $-6$ and $+6$ the predicted minor groove width in A/T flanked GBSs is not only narrower than the rest of the GBS but also narrower than at the corresponding position in G/C flanked GBSs (Fig. 4a and Supplementary Fig. 4A). Importantly, the overall

nucleotide composition (given as A/T content in Fig. 4a) of the GBS and its surrounding region was comparable for the two groups of sequences, indicating that the effect on the two neighbouring nucleotides is a consequence of changing the sequence of the proximal flanks. We also predicted the propeller twist for the same sets of A/T and G/C flanked GBSs and found that the propeller twist differs between the two groups of sequences, especially at positions $-8$ and $+8$ (proximal flanks) (Supplementary Fig. 4B). Next, we repeated the DNA shape prediction for individual GBSs, tested previously in the luciferase reporter assays (Fig. 4b). Since the first half-site (positions $-7$ to $-2$) is identical in all tested GBSs it is not surprising that all GBSs have a similar minor groove width at these positions. Notably, minor groove width of the spacer varies among GBSs, consistent with the known role of the spacer in modulating GR activity[13]. Here we focus on the proximal flank of the second half-site (positions 6–8). For both Cgt and Sgk GBSs, the minor groove width at the flanking position $+8$ is slightly narrower in the G/C flanked version than in the A/T flanked version. In contrast, the neighbouring positions $+6$ and $+7$ exhibit a narrower minor groove width in A/T flanked versions. This result suggests that the crucial structural DNA shape change occurs at positions $+6$, $+7$ and $+8$. For the Pal and FKBP5-1 GBS variants (which do not exhibit a flank effect) the minor groove width is already quite narrow at these positions perhaps explaining why these GBSs do not exhibit a flank effect.

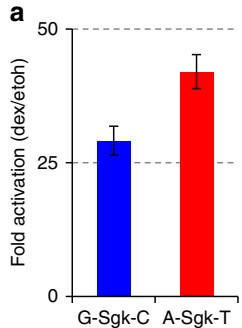

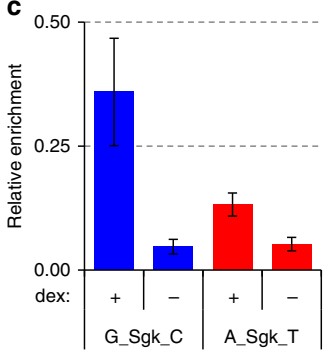

| GBS | Flank | $K_d$ (µM) | s.d. |
|-----|-------|-----------|------|
| Sgk | G/C | 0.55 | 0.07 |
| | A/T | 0.70 | 0.3 |
| Cgt | G/C | 0.98 | 0.09 |
| | A/T | 1.09 | 0.12 |

**Figure 3 | Effect of flanking sites on binding and on regulation in a genomic context.** (**a**) Transcriptional activation of the targeted integrated luciferase reporters with Sgk GBS flanked by either A/T or G/C nucleotides. Average fold induction of the luciferase reporter gene on 1 µM dexamethasone (dex) treatment relative to ethanol (etoh) vehicle ± s.e.m. ($n \geq 3$) is shown. (**b**) Table of EMSA-derived DNA-binding constant ($K_D$) for Sgk and Cgt GBSs with flanking sequences as indicated. S.d. from three independent replicates. (**c**) GR occupancy levels for integrated Sgk-GBS reporters with flanks as indicated was analysed by chromatin immunoprecipitation followed by qPCR for cells treated with either dex (1 µM, 1.5 h) or ethanol as vehicle control. Average relative enrichment at the GBS locus ± s.d. for three clonal lines and three independent replicates is shown.

**GBS flanking nucleotides affect GR-DBD conformation.** Overall, the predicted changes in DNA structure induced by the flanking nucleotides suggest that DNA shape may serve as an input signal that regulates GR activity. To determine if the flanking nucleotides influence GR structure and/or dynamics, we probed the DBD of GR in complex with flank-site Cgt variants by two-dimensional nuclear magnetic resonance (2D NMR) spectroscopy experiments in which nuclei of protein backbone amines ($^1$H, $^{15}$N) are correlated. The resulting spectra provide one signal for each amide and depict the so-called protein fingerprint region, which is unique for each protein construct and chemical (for example, binding-dependent) environment. As expected, addition of proximal flank Cgt variants resulted in spectral changes when compared with unbound DBD (Supplementary Fig. 5A). When we compared the spectra of the complexes between GR DBD and G/C and A/T flanked Cgt oligonucleotides, we found a number of

differences between spectra (Supplementary Fig. 5B). To study these differences in more detail, we analysed the normalized chemical shift perturbation (CSP) data for each residue as described previously[13]. Interestingly, we do not only observe affected amino acid residues in direct vicinity of the altered base pair but rather affected residues reside throughout the whole DBD indicative of global changes in DBD conformation induced by the proximal flanks[20–22] (Figs 5 and 6a).

Next, we selectively changed the flanks at either the 'perfect' half-site (chain A) or at the 'imperfect' half-site (chain B), which is mainly responsible for the flank effect. These experiments showed that changing the flanking nucleotides of the imperfect half-site (AT/AC; Figs 5 and 6b, Supplementary Fig. 5C), resulted in CSPs for several residues (T456, R488, N497, N506, K511). Similarly, changing the proximal flank of the perfect half-site (GC/AC, Fig. 5, Supplementary Fig. 5D) induced peak shifts for multiple residues. Interestingly, however, the residues affected overlapped for some residues (T456 and Y497), whereas they were flank-specific for others (Fig. 5).

As a general rule, NMR spectroscopy is not able to distinguish oligomers with similar conformations or dynamics from one another. During the assignment and CSP calculation though, it became apparent that several residues, which map predominantly to the DNA-recognition helix 1 (G458, C460 and K461), show split peaks meaning more than one signal for a given DBD amino acid (Supplementary Fig. 6). Notably, split peaks were not observed for all residues (example shown for Q520, Supplementary Fig. 6) and a comparison of apo and DNA-bound GR DBD spectra (Supplementary Fig 5a) showed that the extra peaks are not a simple consequence of having a fraction of GR DBD in our samples that is not DNA-bound. Splitted peak patterns are characteristic for either conformational exchange within each monomer or different chemical environments (that is, conformations or DNA sequence) of the individual monomers within the ternary DNA/DBD complex. Observation of a third peak for C460 on substitution of A/T by G/C nucleotides at the proximal flank positions indicates the possible presence of two distinct conformations for one of the individual monomers.

Helix 1 sits in the major groove opposite to the minor groove at positions ($-6$, $-7/+6$, $+7$) where the flanking nucleotides induce a narrowing of that groove. Consequently, the DBD of GR might contact DNA differently, for example, by contacting other nucleotide positions when we change the sequence of the flanks. To test this, we analysed the protein–DNA complex again by NMR spectroscopy but this time by not observing the resonances of the protein but those of the DNA itself. We assigned the imino protons in the 1D spectra for Cgt flanked by either A/T or G/C nucleotides (Supplementary Fig. 7A) and titrated both oligonucleotides with increasing amounts of protein to determine whether the proximal flanks influence protein–DNA contacts within the complex (Supplementary Fig. 7). Consistent with the crystal structure of the GR–DNA complex, these experiments indicate that the DBD contacts both half-sites of the motif at positions $-6$ (G6), $-4$ (T41), $-3$ (G40) or $+2$ (T14), $+4$ (T16). On protein addition, we observed a progressive uniform line broadening for both DNAs, indicative of similar Kd values, which is in agreement with EMSA experiments. When we compared the base pairs contacted between A/T and G/C flanked DNAs, the same set of residues showed evidence for binding to the DBD of GR. However, the imino proton of G46 (position $-9$), whose resonance is well-resolved, led to a more pronounced broadening in the case of the A/T-flanked DNA (Supplementary Fig. 7). This base pair located outside the 15-bp consensus sequence interacts with the DBD of GR, in agreement with contacts formed by helix 3 in the crystal structure[10].

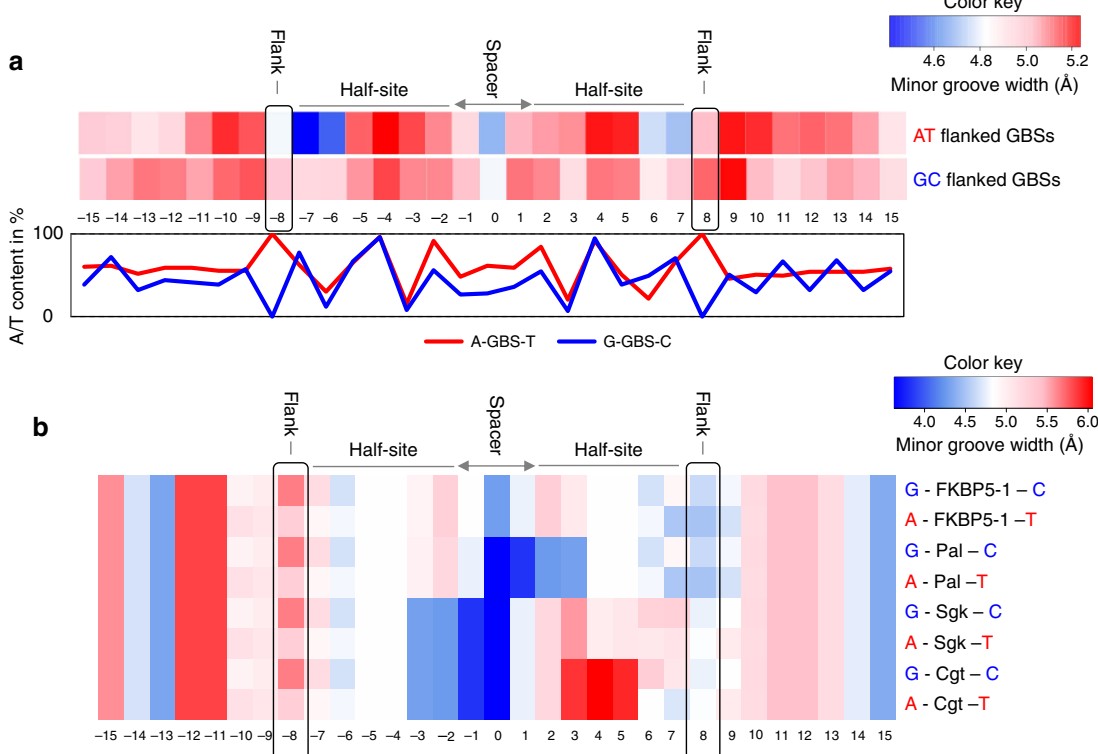

**Figure 4 | Effect of flanking sequences on predicted DNA shape. (a)** Top: predicted mean minor groove width (MGW) for individual nucleotide positions for group of A/T flanked GBSs associated with strong responder genes (83 GBS) and for group of G/C flanked GBSs associated with weak responder genes (75 GBS). Bottom: A/T content (%) at each position for A/T (red) and G/C (blue) flanked sequences used for the analysis. **(b)** Predicted minor groove width for individual nucleotide positions for different GBSs flanked by either G/C or A/T nucleotides.

This highlights a very subtle difference introduced by the flanking nucleotides on the protein–DNA complexes.

Together, our approaches probing changes in the structure indicate that a G/C flank induces several changes in the DBD of GR compared with the GBS with an A/T flank.

**Flank effect requires an intact dimer interface.** To investigate how the DBD of GR might recognize the shape of DNA to modulate GR activity, we tested the role of several candidate residues of the DBD that contact the DNA. As candidates we chose R510, which is part of helix 3 and contacts the flanking nucleotide directly according to the crystal structure[11]. Similarly, K511 might contact the flanking nucleotide and thus shows a significant chemical shift in our NMR experiments on changing the flanks (Fig. 5). In addition, we tested K461 and K465, which reside in the DNA recognition helix 1. Based on the crystal structure, K461 makes a base-specific contact with the G at position $-6/+6$ in the major groove opposite to the position where the flank induces a change in minor groove width, whereas K465 contacts the phosphate backbone[11,23]. When we mutated R510, K511 or K465 to alanine, the flank effect was still observed arguing against a role of these residues in 'reading' the DNA to modulate GR activity (Fig. 7a). Mutating K461 to an alanine resulted in a marked decrease in GR-dependent activation for the A/T and a slight decrease for the G/C-flanked GBS, consistent with decreased activity found for this mutant in previous studies[24]. Interestingly, however, there was still some residual activity for the G/C flanked GBS, the one with the slightly higher affinity (Fig. 3b), whereas no activation was seen for the A/T-flanked variant, which is more active for wild-type GR (Fig. 7a). Interpretation of this result is complicated by the fact that mutating this charged residue alters the binding energetics

and potentially structure of the complex. None the less, our findings suggests that the K461 residue might play a role in interpreting the proximal-flank-encoded instructions and corroborates previous studies[24] that uncovered a role of this residue in interpreting the signalling information provided by GR response elements.

Prior studies have shown that an intact dimer interface is required to read DNA shape and to direct sequence-specific GR activity when changing nucleotides of either the spacer or of GR half sites[13]. Comparison of the binding affinity for A/T- and G/C-flanked Cgt showed that GR's affinity was comparable for both sequences for both wild type (Fig. 3b) and also for A477T DBD (A/T: $3.1 \pm 0.4\,\mu M$; G/C: $3.5 \pm 1.1\,\mu M$) although the affinity was lower for the mutant. To test if the dimer interface plays a role in mediating the flank effect, we tested the impact of disrupting the dimerization interface on proximal-flank-induced modulation of GR activity. As reported previously, mutating A477 of the dimer interface resulted in GBS-specific effects[13]. For the A/T-flanked GBSs Cgt and Sgk, the difference in GR activity between wild type and the A477T mutant was small (Sgk: 8% decrease; Cgt 13% increase, Fig. 7b). In contrast, for the flank with the lower activity, G/C, the A477T mutation resulted in a more pronounced increase in activity for both GBSs tested (Sgk: 50% increase; Cgt: 69% increase, Fig. 7b). Consequently, the difference in activity between the A/T and G/C flanked versions of Cgt and Sgk is smaller for the dimer mutant than for wild type GR (Fig. 7b), indicating that the dimerization domain is involved in transmitting the flank effect. Strikingly, the dimerization interface lies on the opposite side of the GR monomer relative to the flanking nucleotide position, suggesting that a more global change in GR conformation induced by the flanking nucleotides may occur.

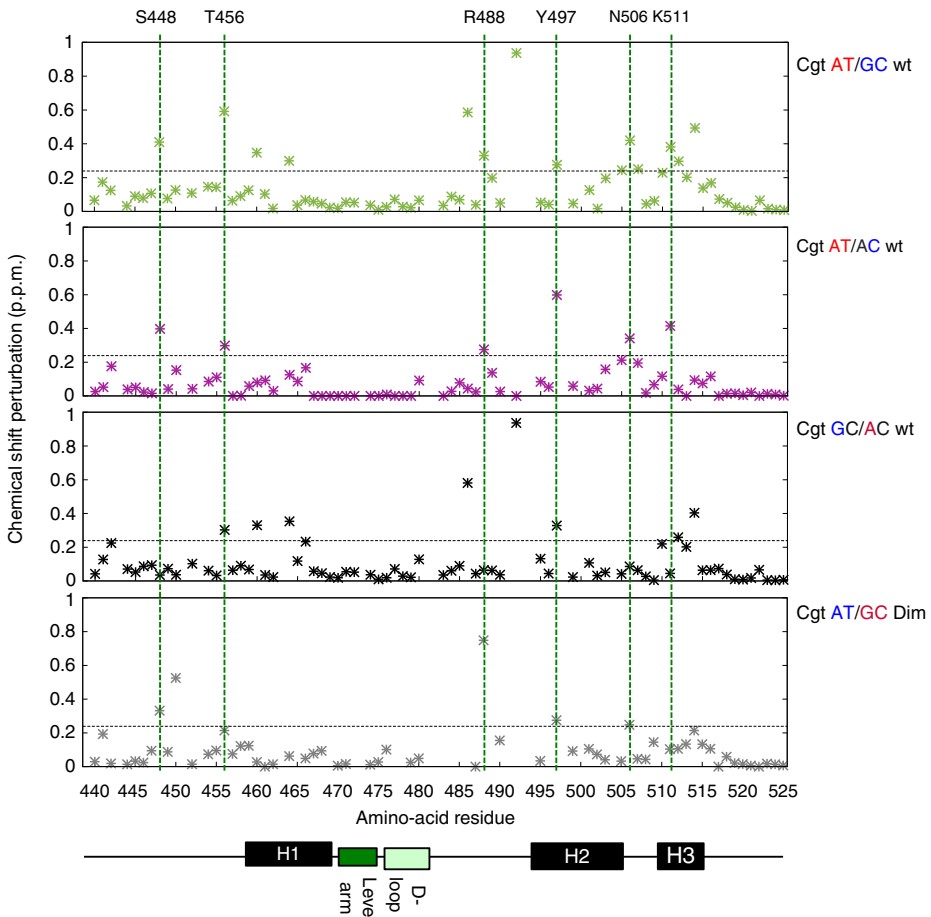

**Figure 5 | NMR chemical shift difference analysis between GBSs with different flanks.** Chemical shift difference of spectra between (top three panels) Cgt flanked by A/T versus G/C; A/T versus A/C; G/C versus A/C for wild-type DBD and (bottom panel) Cgt A/T versus G/C for the dimer mutant DBD (A477T). Horizontal dashed grey lines indicate significance cut-off (average +1 s.d.). Green dashed lines demark amino acid residues with significant shifts when comparing the A/T and A/C sequences.

To further elucidate the role of the dimer interface in transmitting the flank effect, we studied the impact of the A477T mutation on proximal-flank induced conformational changes of GR by 2D NMR spectroscopy (Supplementary Fig. 5E,F). This analysis uncovered two main results. First, several of the residues with significant CSPs for wild type (C460, F464, M505, L507, R511, T512, K514) no longer show a significant shift when we compare the G/C and A/T flanked Cgt for the A477T mutant (Fig. 5, Supplementary Fig. 5F). Second, several peaks that show flank-specific patterns of peak splitting for wild type GR (for example, C460) show an overlapping single peak for the mutated A477T DBD (Fig. 7c). This indicates that proximal flanks can only induce alternative conformations of the DBD when the dimerization interface is intact. Together, these functional and structural analyses of the consequences of disrupting the dimer interface, argue for its role in facilitating flank-induced changes in GR conformation and activity.

## Discussion

Specific recognition of DNA sequences by TFs is a consequence of both base readout and shape readout of the DNA-binding site[25]. In addition to specifying which genes are regulated by a particular TF, the binding site sequence can also play a role in fine-tuning the expression level of genes. For example, binding sites might be able to modulate gene expression as a consequence of differences

in affinity[12,26–28], where high affinity binding sites induce a higher level of transcriptional activation than low affinity binding sites. However, in vitro affinity and in vivo activity often do not correlate[11,29–31]. Accordingly, we find in this study that sequences flanking the core GBS induce changes in activity without apparent changes in affinity derived from in vitro binding studies. One explanation for this apparent disconnect between binding affinity and activity could be that in vitro binding affinity does not reflect binding affinity in vivo. Yet, here we also fail to see a correlation when we compare in vivo occupancy derived from ChIP experiments as a proxy for in vivo affinity. We would like to point out that the interpretation of quantitative comparisons of ChIP efficiencies between binding sites is complicated by possible sequence-specific efficiencies of formaldehyde cross-linking[32]. In this study, we focused on the first flanking nucleotide or 'proximal flank'. However, when we changed the second flanking position, we found an even more dramatic effect, where depending on the sequence of this position GR could either robustly activate transcription, or completely lack the ability to activate transcription (Supplementary Fig. 8A). Again, the modulation of GR activity appears independent of binding affinity, and could be a consequence of conformational changes of the DNA (Supplementary Fig. 8B,C). Together, these findings argue that GBSs can modulate GR activity downstream of binding.

Structural studies[8,11,13], including those presented here, indicate that GBS variants with distinct transcriptional activities

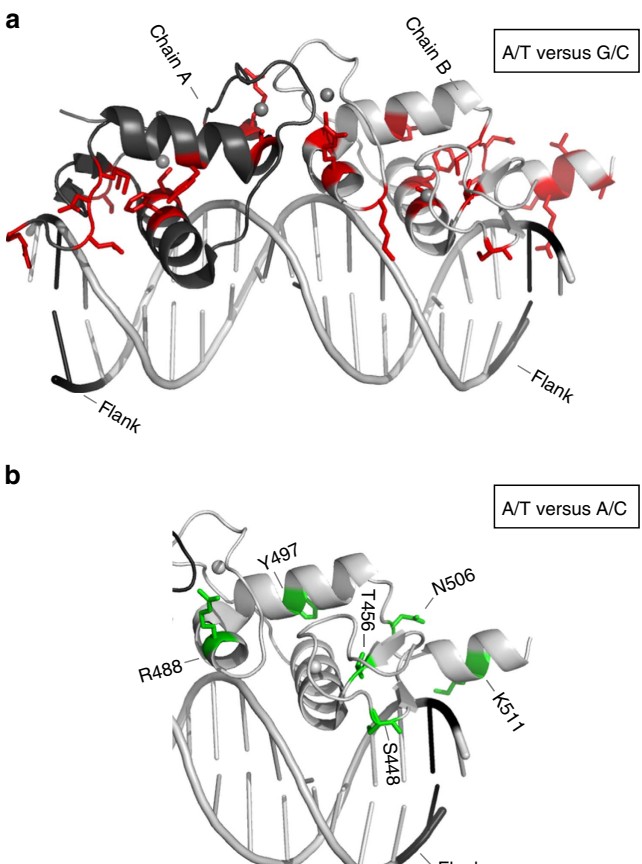

**Figure 6 | Influence of flanking nucleotides on GR structure.** (**a**) Side view of GR DBD crystal structure (PDB: 3G9J) with chains A and B corresponding to each monomer. Amino-acid residues with significant combined [1]H and [15]N chemical shift differences between A/T- and G/C-flanked Cgt sequences are projected onto this GR DBD structure and coloured in red. (**b**) Side view of GR DBD crystal structure with amino acid residues with significant combined [1]H and [15]N chemical shift differences between A/T and A/C flanked Cgt sequence projected in green onto the GR DBD, chain B.

induce alternative conformations in the DBD of GR. These structural changes can be induced by changing the sequence of the spacer, of the half-sites, or as we show here of the nucleotides flanking the core-binding site. Based on the structure, the side chain of R510 and K511 can contact the flanking nucleotides and thus serve as potential 'readers' that interpret the DNA-encoded instructions and translate these into changes in activity. However, when we change these residues to alanines, the flank effect is still observed. This suggests that direct contacts with the flanking nucleotides are not responsible for the flank effect. Instead, the effects of the proximal flanks might be a consequence of the predicted changes in DNA shape. DNA shape, in turn, could induce structural changes in the associated GR dimer partners. To further understand the molecular basis that gives rise to the aforementioned split peaks in our NMR spectra, we turned to molecular dynamics (MD) to simulate how changing the flanks influences the individual monomers. When we compared the overall trajectories, however, we did not observe significant structural differences for either chain A or chain B when we compared the root mean squared deviation (r.m.s.d.) values between the A/T- and the G/C-flanked Cgt GBS. Similarly, we only observed subtle changes when we compared the root mean squared fluctuation (r.m.s.f.) (Supplementary Fig. 9), a measure of

flexibility of the DBD, between the two Cgt flank variants. The changes that do occur, predominantly map to residues at the dimerization interface (Supplementary Fig 10A). In addition, the r.m.s.f. values for monomer B when bound to the G/C-flanked GBS show higher values than those observed for the A/T counterpart indicating that chain B's interaction with the DNA for this sequence is more dynamic (Supplementary Figs 10A and 9). Finally, we compared the median GR-DBD structures (computed from the last 50 ns of the MD simulations) when bound to A/T- or G/C-flanked Cgt. Again, the deviations between these two structures are only small except for the lever arm, which connects the dimerization interface with the DNA recognition helix (Supplementary Fig. 10B). Interestingly, however, changing the flanking nucleotides appears to result in a different relative positioning of the dimer-halves as can be seen from the median conformations for both flank-variants when aligned on chain A (Supplementary Fig. 10C).

Together, our structural approaches showed flank-induced changes in the dynamics and conformation of the dimer partners and in the relative positioning of GR dimer halves. Consistent with previous studies[13], we find that GR's ability to 'read' DNA-shape encoded instructions, in this case as a consequence of changing the flanks, requires an intact dimer interface. Importantly, the mutation in the dimerization domain we studied (A477T) does not result in an inability of GR to dimerize *in vivo*[33]. Therefore, our interpretation of the effect of mutating the dimerization interface is that they are a consequence of perturbing an interface important for communication between dimerization partners or for communication between different GR domains of each monomer, rather than a consequence of an inability of the mutant to bind DNA as a dimer. We find that mutating the dimer interface diminishes flank-induced changes in both GR structure and activity. This suggests that the dimer interface prevents the monomers from adopting an optimal positioning in the major groove and consequently the dimer partners switch between different conformational states to accommodate conflicting optimal contacts at the dimer interface and those with the DNA (Fig. 7d). This might also explain the high degree of flexibility that the dimer interface and connected lever arm display based on the r.m.s.f. values of the MD experiments (Supplementary Figs 9 and 10). Mutation of the dimer interface might release this stress and allow optimal positioning of both dimer partners for contacting the DNA in the major groove. Similarly, conflicts in the optimal positioning of dimer halves might be relieved when mutating K461, which weakens the interactions between DNA and protein[24] thus favouring optimal positioning of the GR partners for interactions at the dimerization interface. To link the structural changes to variations in transcriptional output, we propose that DNA-shape-induced effects on the conformation, dynamics or relative positioning of GR partners influence its interactions with co-regulators by making or breaking interaction surfaces to ultimately modulate the recruitment or activity of the RNA polymerase machinery.

In addition to fine-tuning the activity of TFs, DNA shape also enables paralogous TFs to have distinct DNA-binding preferences[34,35]. For example, members of the Hox family of TFs share a similar consensus recognition sequence, yet have distinct functions *in vivo*. This specificity was explained by Hox-specific DNA shape preferences which enabled the exchange of binding site preferences from one Hox protein to another by swapping shape-recognizing residues[34]. In addition, several other studies have shown a role of nucleotides flanking the core-binding site in guiding TFs to their cognate binding sites[35,36]. For GR, several related nuclear receptors share the same DNA binding specificity *in vitro* yet regulate different

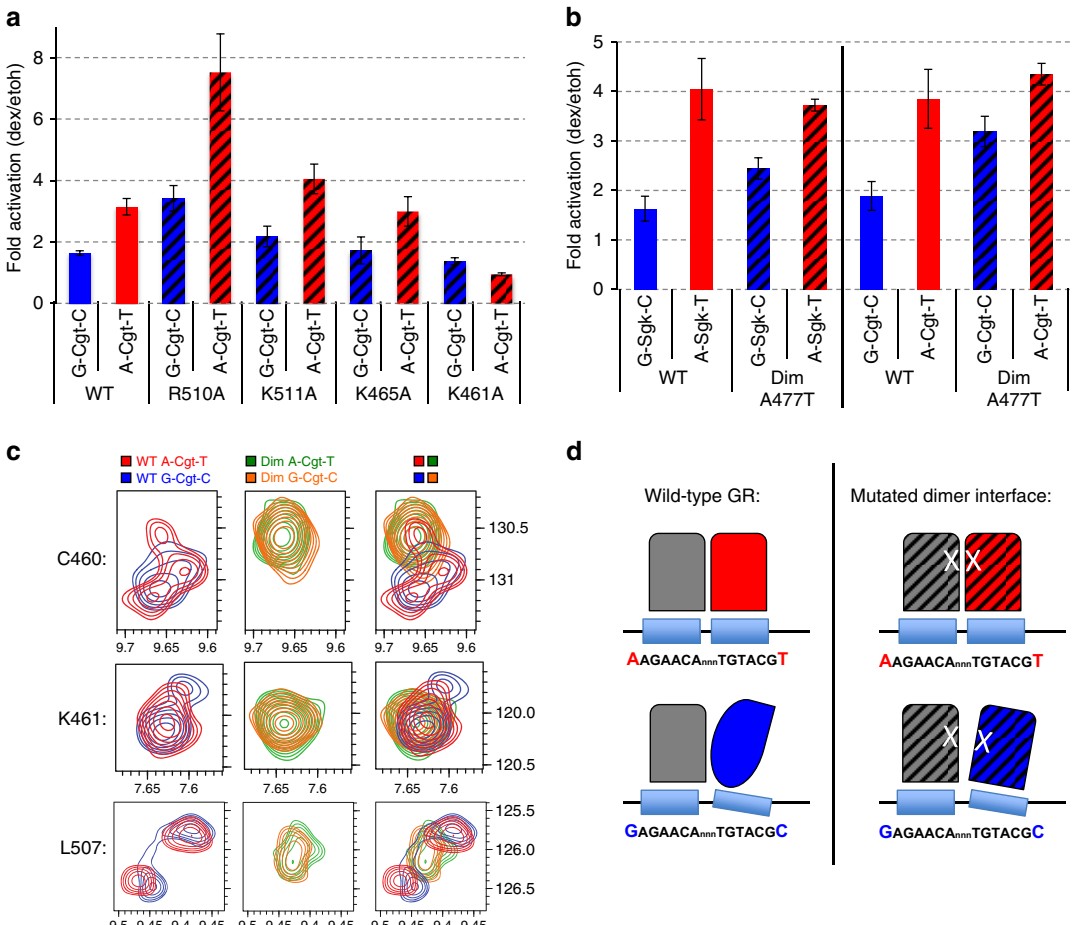

**Figure 7 | Flank effect requires an intact dimer interface.** (**a**) Comparison of transcriptional activation of transiently transfected reporters with GBS as indicated flanked by either A/T or G/C sequences between GR wild-type (WT) and GR variants R510A, K511A, K465A and K461A, respectively. Average induction on 1 μM dexamethasone (dex) treatment relative to ethanol (etoh) vehicle ± s.e.m. ($n \geq 3$) is shown. (**b**) Same as **a** comparing GR wild-type (WT) and dimer mutant (Dim, A477T). (**c**) Zoom-in of $^{1}$H-$^{15}$N-SOFAST-HMQC spectra of selected peaks for residues that show peak-splitting and non-overlapping spectra when comparing GR DBD in complex with either A/T- or G/C-flanked Cgt sequences for (left) wild-type, (middle) A477T dimer mutant DBD and (right) overlay of wild-type and dimer mutant. (Stoichiometry DNA:DBD; 1:2). (**d**) Cartoon depicting (left) how the bases flanking the GBS influence the structure and relative positioning of GR half-sites and (right) how disruption of the dimer interface weakens the effect of the flank on GR structure (and activity).

physiological processes. For example, the androgen receptor promotes myogenesis[37] whereas chronic GR activation results in muscle wasting[38]. We speculate that DNA shape could also generate specificity for this family of TFs by modulation of TF activity downstream of binding. In this scenario, two TFs might bind at the same target site, yet only one adopts an activation-competent conformation. The possibility that only certain binding events induce activation-competent conformations could also explain, in part, why only a minority of genes show changes in their expression level on binding of TFs to regulatory sequences nearby[39].

The activity of TFs towards individual target genes can be modulated by a variety of mechanisms other than the sequence identity of the binding site. For example, a recent study showed that the number of occupied NF-κB-binding sites associated with a gene correlates with the magnitude of activation[39]. However, in addition to being expressed at higher levels, genes with multiple TF-binding sites might display a greater degree of cell-to-cell variability (transcriptional noise) of gene expression[40]. Therefore, we speculate that it could be beneficial for some GR target genes to be under control of a single, highly-active GBS with little transcriptional noise rather than multiple GBSs which induce

greater noise. Another benefit of modulating activity by DNA-shape-induced conformational changes is that this might allow GR to induce different expression levels of a gene from the same binding site depending on its cellular context. This could, for example, occur when a GBS-induced conformation facilitates interaction with a particular co-regulator that is expressed in a cell-type-specific manner. This would be one of several mechanisms that GR can exploit to extract context information from its cellular environment to allow fine-tuning of its activity towards distinct sets of target genes responsible for GR's role in diverse physiological processes including metabolism, inflammatory response and emotional behaviour.

The present study advances our understanding of GBS-mediated regulation of GR activity in several ways. First, we show for the first time that GBSs can modulate GR activity in a genomic context and our *in vivo* occupancy studies indicate that this modulation occurs downstream of binding. Structural studies indicate that this modulation may be a consequence of GBS-dependent conformational changes of individual monomers and of changes in the relative positioning of dimeric partners. Studies with related hormone receptors that heterodimerize have shown allosteric communication between dimerization partners across

the dimerization interface to fine-tune the structure and activity of the complex[41]. Here we propose that GR monomers can change their shape and that the homodimerization partners can change their relative positioning to assemble multiple distinct complexes, effectively allowing a kind of combinatorial regulation of transcriptional output by a single TF. Whether GBSs indeed play a role in modulating the activity of GR towards endogenous GR target genes is still unclear. Arguing in favour of this possibility, we show that GBS sequence features found at GR-bound regions in the genome, specifically the nucleotides flanking the core GBS, show different preferences depending on strength of regulation of the nearby gene. The next step to study the role of GBS composition in the modulation of endogenous target gene expression would be to test the consequences of changing the sequence identity of endogenous binding sites, which, given the recent advances in the ability to edit the genome, has now become within reach.

## Methods

**Plasmids.** Luciferase reporter constructs were generated by inserting a GBS of interest (Supplementary Table 1) by ligating oligonucleotides with overhangs to facilitate direct cloning into the KpnI and XhoI sites of pGL3 promoter (Promega). Mutations of the second flank position (Supplementary Fig. 8) were introduced by site-directed mutagenesis (oligos listed in Supplementary Table 2). Expression constructs for wild-type rat GR, GR dim mutant (A477T) and GR R510A mutant have been described previously[11]. GR mutants K465A and K511A were generated by site-directed mutagenesis (oligos listed in Supplementary Table 2).

Constructs expressing ZFNs against the *AAVS1* locus have been described elsewhere[14,42]. Donor constructs for luciferase reporter addition to the *AAVS1* locus were assembled as described[14]. The donor constructs consisted of regions of homology flanking the position where the ZFNs induce the double strand break, a promoter-less GFP gene and the GBS sequence as indicated upstream of a minimal SV40 promoter driving expression of the firefly luciferase gene derived from the pGL3-promoter plasmid (Promega).

**Cell lines, transient transfections and luciferase assays.** U2OS (ATCC HTB-96) and U2OS cells stably transfected with rat GRα[43,44] were grown in DMEM supplemented with 5% FBS. Transient transfections were done essentially as described[11]. Luciferase activity was measured using the dual luciferase assay kit (Promega).

**Electrophoretic mobility shift assays.** EMSAs were performed as described previously[15]. Briefly, a series of GR DBD dilutions were mixed with $1.25 \times 10^{-9}$ M DNA (oligos listed in Supplementary Table 3) in 20 mM Tris pH 7.5, 2 mM MgCl2, 1 mM EDTA, 10% glycerol, 0.3 mg ml$^{-1}$ BSA, 4 mM DTT, 0.05 µg µl$^{-1}$ dIdC. Reaction mixes were incubated for 30 min to reach equilibrium, loaded onto running native gels and scanned using a FLA 5,100 scanner (Fujifilm) to quantify free $[D]$ versus total $[D]t$ DNA. Equilibrium binding constants ($K_D$) were determined by non-linear least squares fitting of the free protein concentration $[P]$ versus the fraction of DNA bound ($[PD]/[D]t$) to the equation $[PD]/[D]t = 1/(1 + (K_D/[P]))$.

**Targeted Integration of GBS reporters.** Cell lines with stably integrated GBS reporters were isolated as described previously[14]. Briefly, cells were transformed with ZFN and donor construct by nucleofection (Amaxa), GFP-positive pools of cells were isolated by flow-activated cell sorting (FACS) and single-cell-derived clonal lines were isolated. To identify clones with a correct integration of the donor construct at the *AAVS1* locus, 40 ng of chromosomal DNA was analysed by PCR using a primer targeting the donor construct (Luc-fw: 5′-Tcaaagaggcgaactgtgtg-3′) and a primer targeting the genomic *AAVS1* locus that directly flanks the site of integration (R5: 5′-ctgggataccccgaagagtg-3′)(Fig. 1a and Supplementary Fig. 1A).

**Chromatin immunoprecipitation.** ChIP assays were performed as described using the N499 GR-antibody[15]. For each ChIP assay, approximately five million cells were treated with 0.1% ethanol vehicle or 1 µM dexamethasone for 1.5 h. Primers used for quantitative PCR (qPCR) are listed in Supplementary Table 4.

**RNA isolation and analysis by qPCR.** RNA was isolated from cells treated for 8 h with 1 µM dexamethasone or with 0.1% ethanol vehicle using the RNeasy mini kit (Qiagen). The Turbo DNA-free kit (Ambion) was used to remove trace amounts of contaminating chromosomal DNA prior to reverse transcription using random primers and 500 ng of total RNA as input. Resulting cDNA was analysed by qPCR using *Rpl19* as an internal control for normalization. Primers used are listed in Supplementary Table 4.

**Computational analysis of ChIP-seq and gene expression data.** Microarray data sets in U2OS cells were taken from ref. 15 (E-GEOD-38971). ChIP-seq data sets from the same study were downloaded as processed peaks from GEO (E-MTAB-2731). The differentially expressed (adjusted $P$ value < 0.05) genes in U2OS cells were assigned to two different groups. The first group consisted of the 20% most upregulated genes on hormone treatment (log2-fold change dexamethasone/ethanol vehicle ranging from 1.91 to 7.86; 290 of 1,447 genes). Next, we extracted the ChIP-seq peaks falling in a 40 kb window centred on the transcription start site of each gene (543 peaks in total from 290 genes of this group). For comparison, we extracted a similar number of peaks (532) from genes (688) showing only weak regulation (absolute log2-fold change ≤ |0.72|). For each group of peaks, we performed *de novo* motif discovery using RSAT peak motifs (default settings, including dyad-analysis algorithm and the TRANSFAC version 2010.1 motif collection)[16]. Peak motifs automatically compare detected motifs to annotated motif collections, and motifs matching the GR consensus motif (depicted in Fig. 2a) were manually extracted.

To compare ChIP-seq peak heights between GR-bound regions harbouring either A/T or G/C flanked GBSs, GR peaks were first scanned for the occurrence of a GBS-match with RSAT matrix scan (Transfac matrix M00205, $P$ value cut-off: $10^{-4}$ refs 16,45)). Next, peaks were grouped according to the sequence of the flanks (A/T versus G/C) and median peak height was calculated to produce Supplementary Fig. 3.

To score the enrichment of A/T GBS and G/C-flanked GBSs in the peaks associated with strong and weak upregulation, respectively (Supplementary Fig. 2), RSAT matrix-quality was used to compute normalized weight differences (NWD)[46]. The input motifs for matrix-quality were derived from the above-mentioned matrices corresponding to GR motifs found with peak motifs, enforcing only A/T or G/C at the flank position.

**DNA shape prediction.** For DNA shape prediction, we used GBSs associated with weakly and strongly responsive GR target genes. For the weak and strong peak data sets, we extracted the sequence of all GBSs flanked by either G and C (75 GBSs) or A and T (83 GBSs), respectively. The sequences were aligned based on the GBS spacer by setting the centre spacer position to 0. Minor groove width and propeller twist were derived for each position in the aligned sequences using a high-throughput DNA shape prediction approach[19]. To test for differences in DNA shape features between the weak and strong peaks, Wilcoxon test $P$ values were calculated for each nucleotide position separately.

**NMR.** *Protein expression and purification*. [15]N-labelled wild-type and A477T mutant rat GR DBD (residues 440–525) were expressed and purified essentially as described previously[13] except that a codon optimized construct for expression in *Escherichia coli* was used here. In brief, proteins were expressed in *E. coli* (T7 Express; NEB) using the pET expression system in M9 minimal medium[47]. Expression was induced at an $OD_{600}$ of 0.6–0.9 using 0.25 mM IPTG (Amresco). Temperature was lowered from 37 to 25 °C on addition of IPTG and cultures grown overnight. Cells were harvested and lysed followed by protein separation by IMAC and IEX chromatography. The latter was done after extensive dialysis against salt-free buffer. Final dialysis at the end of protein purification was carried out against NMR buffer (20 mM sodium phosphate; 100 mM NaCl; 1 mM DTT; pH 6.7).

*Protein–DNA complex formation*. Single-stranded DNA oligos (salt-free and lyophilized) were purchased from MWG and purified as described[13]. Buffer was exchanged to water using NAP10 gravity flow columns (GE Healthcare) and annealed according to a standard protocol. Success of annealing was evaluated using proton-detected 1D NMR spectra. Protein–DNA complexes for 2D NMR were prepared essentially as described[13] by mixing protein solution of either GRα or GRα-dim in onefold NMR buffer with dsDNA oligos. Final concentrations of protein and DNA was 40 µM and 53 µM, respectively, resulting in a molar ratio of 1:1.33. Samples were supplemented with 5% $D_2O$ the lock. Water and twofold NMR buffer was added to give a final sample volume of 500 µl. Sequence of oligos is described in Supplementary Table 5.

*NMR and CSP analysis*. [1]H-[15]N-HSQC spectra were recorded as SOFAST versions[48] at 35 °C on a Bruker AV 600 MHz spectrometer (Bruker, Karlsruhe, Germany) equipped with a cryo-probehead. TopSpin (version 3.1, Bruker) was used for data processing, including zero filling and linear prediction. The transfer of previous assignment[13] and general data evaluation were done using the CCPN software package (version 2.1.5)[49].

CSP was calculated using the following formula[50]:

$$\sqrt{\left(^1H[p.p.m.]\right)^2 + \left(^{15}N[p.p.m.] \cdot \gamma_{15_N}/\gamma_{1_H}\right)^2}$$

where [1]H and [15]N refer to the mathematical difference of individual hydrogen and nitrogen chemical shifts of two distinct peak maxima. Gyromagnetic ratio ($\gamma_i$) of nuclei $i$, where i is [1]H or [15]N, is used for normalization.

*DNA assignment*. NMR experiments were recorded at 700 MHz on an Avance III Bruker spectrometer equipped with a TCI z-gradient cryoprobe. NMR data were acquired at 15 and 20 °C. Solvent suppression was achieved using the 'Jump and Return' sequence combined to WATERGATE[51–53]. 2D NOESY spectra were acquired with mixing times of 400 and 50 ms. NMR data were processed using

TopSpin and analysed with Sparky software packages (Goddard, T.D. and Kneller, D.G., SPARKY 3, the University of California, San Francisco). [1]H assignments were obtained using standard homonuclear experiments. The resonances found between 10 and 14 p.p.m. are characteristic of protons involved in hydrogen bonds, generally due to the formation of base pairs. The imino proton spectra of A/T- and G/C-flanked DNAs, showed the formation of DNA duplexes. The A:T Watson–Crick base-pairs were discriminated from G:C base-pairs by the strong correlation between the thymine H3 imino proton and the H2 proton of adenine. In a G:C Watson–Crick base-pair, two strong NOEs cross-peaks are observable between the guanine H1 imino proton and the cytosine amino protons. Base-pairings were next established via sequential nuclear Overhauser effects observed in 2D NOESY spectra at different mixing times.

*DNA–protein titration*. Proton detected 1D NMR spectra with double WATERGATE sequence for water suppression[54] were used for titration experiments. Inter-gradient delay of WATERGATE sequence was set to 80 µs to obtain a maximum signal intensity of dsDNA-specific hydrogen bonds at ~12 p.p.m. About 500 µl of 50 µM dsDNA in 1× NMR buffer without protein was used as initial concentration (incl. 5% $D_2O$). Unlabelled GRα (1.2 mM stock concentration in NMR buffer) was added stepwise to achieve DNA–sprotein ratios of 0.25; 0.50; 0.75; 1.00; 1.25; 1.50; 1.75; 2.00; 2.50; 3.00, while minimal dilution of dsDNA occurred (final concentration of dsDNA at 1:3 ratio was 45 µM). All titrations experiments were performed at 25 °C, monitoring the imino protons region of 1D spectra. Intensities of imino protons were measured at each point of the titration. Ratios of intensities between bound-DNA and free-DNA were calculated for both A/T DNA and G/C DNA. All peaks showed similar decreases in intensity with increasing DNA–protein ratios, with the exception of G46 which exhibited a more pronounced broadening in the case of the A/T DNA.

**MD simulations.** *Molecular systems*. Classical MD simulations were carried out for A/T and G/C flank variants of the Cgt GBS. The initial structure was prepared based on a crystal structure of the GR DNA-binding site in complex with the Cgt-binding site (PDB ID 3FYL[11]). Position +5 was mutated *in silico* (C to A). Five and four nucleotides per strand in a perfect B-form were added to the 5′ and 3′ side of the DNA fragment, respectively, resulting in DNA fragments with 24 nucleotides length: 5′-CACCAAGAACATTTTGTACGTCTC-3′ and 5′-CACCGA GAACATTTTGTACGCCTC-3′ for the A/T and G/C Cgt flank variant, respectively.

*Molecular dynamic simulation*. The simulations were performed with the program package NAMD 2.10 (ref. 55) using CHARMM27 force field[56]. The DNA fragments of initial structures were energy minimized (3,000 steps of conjugate gradient) to remove energetically unfavourable conformations resulting from the addition of the additional nucleotides. The systems were solvated in TIP3P water[57] and a total of 35 sodium ions were placed randomly within a minimum distance of 10.5 Å from the solute and 5 Å between sodium ions to ensure a zero net charge for the solute–solvent–counterion complex. The systems contained ~127,000 atoms. The final complexes were equilibrated by 5,000 steps of energy minimization, followed by a 30 ps MD simulation (time step 1 fs) to heat up the system to 300 K by velocity scaling. Next, a relaxation 200 ps (time step 1 fs) was performed for an NPT ensemble. Periodic boundary conditions were implemented with the particle-mesh Ewald method[58] for electrostatic interactions with cut-off distance 14 Å. Lennard–Jones interactions were truncated at 14 Å. The SHAKE algorithm was applied to constraint all bonds involving hydrogen atoms. Three independent, 100-ns-long MD simulations were performed in constant pressure (1 bar) and constant temperature (300 K) with a 2 fs time step for each A/T and G/C flank GBS. During these simulations, pressure and temperature were maintained constant using langevin dynamic barostat and Nosé − Hoover Langevin thermostat. The terminal base pairs of the DNA fragments were restrained harmonically. A simulation run was further prolonged to 300 ns for complexes with both A/T- and G/C-flanking nucleotides.

**Data availability.** Microarray (E-GEOD-38971) and ChIP-seq (E-MTAB-2731) data are deposited in the GEO repository. All other data are available from the corresponding authors upon request.

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

## Acknowledgements

We thank Edda Einfeldt and Katja Borzym for excellent technical support. This work was funded by the DFG (grant ME4154/1-1 to M.J.) and NIH (grant R01GM106056 to R.R.).

## Author contributions

S.S. performed and conceived the experiments and analysed the data. M.J., I.L. and B.K. performed and analysed the NMR experiments. M.B.H. and P.I. performed and analysed the MD simulations. I.D. and R.R. performed and analysed the DNA shape predictions. S.S., M.T.-C., M.V. and S.H.M. designed and supervised the study and wrote the manuscript with input from all authors.

## Additional information

DOI: 10.1038/ncomms13784      OPEN

# Corrigendum: Sequences flanking the core-binding site modulate glucocorticoid receptor structure and activity

Stefanie Schöne, Marcel Jurk, Mahdi Bagherpoor Helabad, Iris Dror, Isabelle Lebars, Bruno Kieffer, Petra Imhof, Remo Rohs, Martin Vingron, Morgane Thomas-Chollier & Sebastiaan H. Meijsing

Nature Communications 7:12621 doi: 10.1038/ncomms12621 (2016); Published 1 Sep 2016; Updated 22 Nov 2016

The financial support for this Article was not fully acknowledged. The Acknowledgements should have included the following:

M.B.H. and P.I. are grateful for computational resources provided by the North-German Supercomputing Alliance and the ZEDAT cluster Soroban of the Freie Universität Berlin.

