## [Peer Review File · Nature Communications]

Reviewer #1 (Remarks to the Author):

The authors show convincingly that GR-dependent gene activation can be modulated by base-pairs flanking the canonical 6 bp glucocorticoid binding sites. They conclude this from measuring GR activity in various genomic contexts and excluding a possible role GR occupancy by CHIP-seq experiments. They also offer a structural interpretation of this observation based on computational analysis of DNA structure, MD simulation of protein-DNA complexes, and NMR experiments, in particular ¹⁵N HSQC or HMQC. At various places in the abstract and main text they call this "structural studies", which is not correct. For instance, differences in peak positions in HSQC spectra indicate changes in chemical environment, but a more detailed structural interpretation is notoriously difficult, in particular in DNA complexes with large charge effects. Thus, in my view, details of the structural interpretation of the flank effect are still speculative and await real structure determinations of complexes by either X-ray crystallography or NMR.

Specific remarks:

- Calculations of DNA shape for naked DNA (p. 10) showed a narrowing of the minor groove for bp's involved in the flank effects. It is not clear how relevant this is in view of the flexibility of DNA, which allows small deformations at low energetic cost.
- As the flanking bp 7 is contacted by the protein, it is not surprising the changing this bp causes multiple changes in HSQC spectra (p.11-12, Fig 5).
- The section on MD simulations (p. 14) is rather inconclusive and can be deleted.
- Mutating charged residues that interact with DNA (p. 15) may drastically alter the binding energetics and structure of the complex. This would obscure any interpretation of the flank effect.
- Similar arguments hold for the A477T mutation in the dimer interface. In view of the remaining GR activity this causes probably only a small repositioning of the monomers rather than a "disruption of the interface" (abstract).
- I would suggest to drastically shorten the so-called structural sections, p. 10-16.

In conclusion I recommend publication after major revision addressing the points listed above.

Reviewer #2 (Remarks to the Author):

In "Sequences flanking the core binding site modulate glucocorticoid receptor structure and activity", Schone and colleagues address a very important topic, namely the mechanisms underlying the regulatory specificity (activation versus repression) of GR for a given gene. The authors used U2OS cells with specific GR binding elements integrated to a designated target "safe harbor" locus in the genome to assess the effects of varying nucleotides immediately flanking the core GR binding 15-mer sequence. In my opinion, this data is the most interesting contribution of the study, although the panel of GREs taken from GR target genes is rather limited. Overall, the authors seem to re-inforce the model whereby GR reads the DNA-encoded signal from individual GREs and adopts a conformation that determines the regulatory outcome. This model has been promoted by the works of Keith Yamamoto and the senior author of this manuscript. However, it will serve the general community better if the paper includes other (even conflicting) reports in the literature for a more objective discussion of the topic. It is fine to promote a less-accepted model (I agree that the authors' model is an intriguing one) by preferential invocation of certain mechanisms that explain their data, but I felt in several places throughout the manuscript that the interpretation of data in the literature is rather subjective and the discussions exclude other important studies. For example, the recently explored controversy about whether the GR dim mutant proteins indeed form monomers in vivo is not mentioned at all (see major comment #5). Below I list specific comments regarding this and other points. This reviewer does not have sufficient knowledge to assess the MD simulations or NMR data analysis, and hence I cannot comment on those sections.

Major comments:

1. The authors should tone down some text where they express their views in statements that sound like universally accepted facts. Examples of such text include page 3 line 87-88, "Depending on the sequence..., the direction of regulation is influenced..." and page 17 line 434 "Binding sites can modulate gene expression by inducing changes in affinity, ...". In the latter instance, the authors cite one study (Bain DL et al. J Mol Biol 2012) where the data seem more of an exception than a rule, given the numerous studies that show little correlation between binding affinity and transcriptional activity.

2. Have the authors checked whether the GBS reporter was integrated to the intended safe harbor locus but not elsewhere in the genome? If there were off-target integrations, the fold induction of the reporter would be contaminated by the GBSs in the other various genomic contexts, which counters the attempt to equalize the genomic context of the GBSs.

3. In Figure 2B-C, it is not stated whether the experiments were done with transiently transfected reporters or the stable clones expressing reporters from the safe harbor locus. Please specify. In Figure 3, are the integrated reporters from regular (random integration) stable lines or from the targeted integration into the safe harbor locus?

4. DNA shape feature analysis seems to have been based on computationally predicted measures. This point is not obvious unless the reader tracks it down to the Methods section. It should be clearly stated in the main text that the analysis was computational and not based on experimental measurements which were specifically generated for the study. The current description gives the impression that the shape analysis produced definitive results, whereas the findings are simply theoretical explorations. Whenever computational analyses and inferences are employed, it should be clearly specified.

5. It is puzzling that there is no mention of the controversy about the A477T (the rat "GR dim" mutant). A recent study used multiple live microscopy techniques to generate convincing evidence that the so-called dimerization mutant is in fact capable of forming dimers in living cells (Presman DM et al. PLoS Biol 2014). Whether the GR mutant proteins exist as dimers or monomers would clearly affect the interpretation of the data presented in this study. What kind of dimerization status is assumed for the GR mutant in a statement such as "mutation of the dimer interface might release the conformational stress"?

6. As the authors mention in the discussion on page 20, it would be interesting to examine not only the transcriptional activity but also the cell-to-cell variability across the GBSs in the same genomic context using the safe harbor reporter clones. The cell-cell variability can be readily measured by quantifying the luciferase signal from individual cells.

7. For the data shown in Figure 7, how were the GR mutant variants introduced into what cells? I couldn't find the description about how these mutants were expressed. If by transient transfection, the results are affected by the highly variable expression level. In addition, depending on the cell context, the endogenous GR can interact with the exogenous mutant GR. What controls were performed to address these issues?

Minor comments:

1. Page 3 line 82-83, "... post-translational modifications of DNA..." needs to be revised.

2. In Figure 2A, I could not find the number of genes in the control group of weak responders in the legend or the main text. In addition, by the control group do the authors mean constitutively expressed genes or induced genes with very low fold change? The intended meaning is not clear.

3. Another incidence of overstating a minority view in the field is seen on page 20 in the paragraph starting in line 509. The authors cite a few selected studies to make general statements that are not widely believed: "...the number of occupied binding sites...correlates with the magnitude of

activation."; "...genes with multiple TF binding sites tend to have a greater degree of cell-to-cell variability..." I suggest that these statements must be rephrased to indicate that there is currently a limited amount of data supporting these ideas.

Reviewer #3 (Remarks to the Author):

This is a beautiful study on the effects of the flanking nucleotides of the binding sites of a transcription factor, here the glucocorticoid receptor (GR). The effects are not negligible. Through a combination of bioinformatic, expression, biochemical, NMR, and molecular dynamics (MD) analyses, the authors are convincingly able to link changes in DNA shape to changes in GR shape and activity. It is primarily the NMR and MD which allow them to suggest that the flanking nucleotides induce subtle conformational changes and changes in the relative positions of the two monomers. These findings are highly relevant to understand not only GR action, but more generally that of other dimeric nuclear receptors and TFs.

Major comments:

- (1) Equal DNA binding: the final mutants shown in Fig. 7B being crucial for the story, the authors should verify that DNA binding affinity/occupancy of the GR mutants is not affected by the flanking nucleotides.
- (2) Are there natural polymorphisms? Can polymorphisms in flanking nucleotides be identified and linked (speculatively or not) with altered GR responses?

Minor comment:

- (3) Abstract: the expression "binds short DNA fragments" does not seem appropriate for the binding mode of a TF within the context of genomic DNA.
- (4) Fig. 2: it should be clearly indicated in the legend that these are transient transfection assays.
- (5) Fig. 2C: this panel is too cryptic; it is difficult to read (notably also because it is not obvious what the wild-type sequences of these sites are).
- (6) Fig. 4: it should be clearer from the legend (title and contents) that the DNA shape is calculated/predicted.

Response to reviewer's comments:

Reviewer #1

Comment 1: "At various places in the abstract and main text they call this "structural studies", which is not correct. For instance, differences in peak positions in HSQC spectra indicate changes in chemical environment, but a more detailed structural interpretation is notoriously difficult, in particular in DNA complexes with large charge effects. Thus, in my view, details of the structural interpretation of the flank effect are still speculative and await real structure determinations of complexes by either X-ray crystallography or NMR."

We understand the concerns raised by reviewer 1 about a pure structural interpretation of the changes observed on HSQC spectra measured for the GR:DNA complexes. However, the distribution of chemical shifts perturbations induced in the protein by changing the flanking nucleotides includes both residues located close to the DNA and at remote positions, in particular at the dimerization interface. This observation supports a global effect on the homodimer rather than a pure charge effect. A compelling example is provided by the cysteine 460 whose correlation peak on the ¹H-¹⁵N HSQC displays several states in both complexes. The distribution of these correlations is clearly different upon changing the sequence identity of the flanking nucleotide, indicating a redistribution of chemical shift environments. We believe that this is due to both structural and dynamical changes of the GR homodimer that may not be captured by a single structure. To account for this comment, we modified the abstract and introduction (p.4) to reduce the emphasis on a purely structural interpretation of our observations in solution. For example, in the abstract we replaced "structural studies" with "experiments with atomic resolution".

Comment 2: Calculations of DNA shape for naked DNA (p. 10) showed a narrowing of the minor groove for bp's involved in the flank effects. It is not clear how relevant this is in view of the flexibility of DNA, which allows small deformations at low energetic cost.

This is generally a valid concern of the reviewer. However, in this specific case flexibility is less of a concern because a narrowing of the minor groove leads to inter-base pair hydrogen bonds in the major groove, which makes this region of the DNA rigid. These stabilizing hydrogen bonds also give rise to a propeller twisting of the bases. Proteins often specifically recognize such rigid elements of narrow minor grooves (see Rohs et al. Nature 2009 for a more detailed discussion) because they are fairly stable compared to regions where low energetic cost leads to a widening of the minor groove.

Comment 3: As the flanking bp 7 is contacted by the protein, it is not surprising the changing this bp causes multiple changes in HSQC spectra (p.11-12, Fig 5).

As we stated in the manuscript, we do not only observe changes in HSQC spectra in the vicinity of the flanking nucleotide, but throughout the whole DBD including the dimer interface indicating global conformational changes. To make it clearer, the revised manuscripts now includes the following sentence (page 12): "Interestingly, we do not only observe affected amino acid residues in direct vicinity

of the altered base-pair but rather, affected residues reside throughout the whole DBD indicative of global changes in DBD conformation induced by the proximal flanks."

Comment 4: The section on MD simulations (p. 14) is rather inconclusive and can be deleted.

We agree with the reviewer that the MD simulation did not uncover major new insights into how the flanking nucleotides influence the structure of individual GR monomers. One interesting, and novel, finding from the MD simulations was that the flanking nucleotides might influence the relative positioning of the dimerization partners. Therefore, instead of removing this section entirely, we decided to shorten it and move it to the discussion (page 18) and to move the figure panels describing the MD results to the supplementary material.

Comment 5: Mutating charged residues that interact with DNA (p. 15) may drastically alter the binding energetics and structure of the complex. This would obscure any interpretation of the flank effect.

This is a valid concern, and to address this we have added a sentence to the section where we discuss the results regarding the K461A mutant (page 15 and see below). Further, we use careful wording (might, suggest) to avoid over-interpretation of our results. ..." Interpretation of this result is complicated by the fact that mutating this charged residue alters the binding energetics and potentially structure of the complex. None the less, our findings suggests that the K461 residue might play a role in interpreting the proximal-flank encoded instructions and corroborates previous studies²⁴ that uncovered a role of this residue in interpreting the signaling information provided by GR response elements. "....

Comment 6: Similar arguments hold for the A477T mutation in the dimer interface. In view of the remaining GR activity this causes probably only a small repositioning of the monomers rather than a "disruption of the interface" (abstract).

This is a very good point, which was also raised by reviewer #2. We agree that the A477T mutation likely does not result in an inability of GR to dimerize. Therefore, we changed this sentence in the abstract from "disruption of the interface" to: "mutating the interface". Further, on page 19, we added a section to refer to a study showing that the A477T mutant can still dimerize *in vivo* and to clarify our interpretation of the effect of mutating the dimerization interface.

Comment 7: I would suggest to drastically shorten the so-called structural sections, p. 10-16.

To address this, we shortened this section, and moved the MD simulations to the discussion.

Reviewer #2:

Major comments:

1. The authors should tone down some text where they express their views in statements that sound like universally accepted facts. Examples of such text include page 3 line 87-88, "Depending on the sequence..., the direction of regulation is influenced..." and page 17 line 434 "Binding sites can modulate gene expression by inducing changes in affinity, ...". In the latter instance, the authors cite one study (Bain DL et al. J Mol Biol 2012) where the data seem more of an exception than a rule, given

the numerous studies that show little correlation between binding affinity and transcriptional activity.

We have revised the above-mentioned sentences to moderate our statements. In addition, we have added several references to papers that indicate that differences in binding affinity can direct differences in transcriptional activity (page 17) to provide a more balanced view. Page 3: ...” Some studies suggests that depending on the sequence of the GR binding sequence (GBS), the direction of regulation might be influenced, i.e. whether GR will activate or repress transcription⁸⁻¹¹.” ... Page 17: ...” For example, binding sites might be able to modulate gene expression as a consequence of differences in affinity^{12,26-28}, where high affinity binding sites induce a higher level of transcriptional activation than low affinity binding sites.” ...

2. Have the authors checked whether the GBS reporter was integrated to the intended safe harbor locus but not elsewhere in the genome? If there were off-target integrations, the fold induction of the reporter would be contaminated by the GBSs in the other various genomic contexts, which counters the attempt to equalize the genomic context of the GBSs.

For each of the clonal lines with integrated GBS reporter, we check for integration at the correct locus using a diagnostic PCR. Further, the data for each of the reporters presented is an average of at least three different clonal lines to assure that the differences in activity we observe are reproducible and not simply a consequence of clonal variation, for example due to additional off-target integrations. The clonal lines showed comparable basal and induced reporter activity arguing against off-target integration, which in our experience is a rare event. This was also found in the original study describing the method we used to integrate reporters at the safe harbor locus, which reported off target integration for <10% of clonal lines examined (Hockemeyer et al 2009).

3. In Figure 2B-C, it is not stated whether the experiments were done with transiently transfected reporters or the stable clones expressing reporters from the safe harbor locus. Please specify. In Figure 3, are the integrated reporters from regular (random integration) stable lines or from the targeted integration into the safe harbor locus?

The figure legend of Figure 2B and 2C now states that experiments were done using transiently transfected reporters. The experiments presented in figure 3 were done using clonal lines with targeted integration of the reporters into the safe harbor locus. This information is now included in the revised figure legend.

4. DNA shape feature analysis seems to have been based on computationally predicted measures. This point is not obvious unless the reader tracks it down to the Methods section. It should be clearly stated in the main text that the analysis was computational and not based on experimental measurements which were specifically generated for the study. The current description gives the impression that the shape analysis produced definitive results, whereas the findings are simply theoretical explorations. Whenever computational analyses and inferences are employed, it should be clearly specified.

The DNA shape data presented are indeed predictions. To make this clearer, we have we changed the legend of figures 4 and S8 to specify that the DNA shape data presented are predictions. Further, we added the following sentence and reference to the main text (page 10):“The DNA shape features were predicted using a high-throughput method that has been extensively validated based on experimental data (Zhou et al. NAR 2013)” This method was extensively validated based on all available X-ray crystallography, NMR spectroscopy, and hydroxyl radical cleavage data.

5. It is puzzling that there is no mention of the controversy about the A477T (the rat "GR dim" mutant). A recent study used multiple live microscopy techniques to generate convincing evidence that the so-called dimerization mutant is in fact capable of forming dimers in living cells (Presman DM et al. PLoS Biol 2014). Whether the GR mutant proteins exist as dimers or monomers would clearly affect the interpretation of the data presented in this study. What kind of dimerization status is assumed for the GR mutant in a statement such as "mutation of the dimer interface might release the conformational stress"?

In our mind the controversy no longer exists as by now several studies, including the one suggested by the reviewer, have busted the myth that the GR dimer mutant cannot dimerize. To clarify our interpretation of the consequences of mutating the dimerization interface, we have changed a sentence in the abstract from ..“disruption of the interface” to: “mutating the interface” ... Further, on page 19, we added a section to refer to the Presman study showing that the A477T mutant can still dimerize *in vivo* and to clarify our interpretation of the effect of mutating the dimerization interface.” Importantly, the mutation in the dimerization domain we studied (A477T) does not result in an inability of GR to dimerize *in vivo*³³. Therefore, our interpretation of the effect of mutating the dimerization interface is that they are a consequence of perturbing an interface important for communication between dimerization partners or for communication between different GR domains of each monomer, rather than a consequence of an inability of the mutant to bind DNA as a dimer.”....

6. As the authors mention in the discussion on page 20, it would be interesting to examine not only the transcriptional activity but also the cell-to-cell variability across the GBSs in the same genomic context using the safe harbor reporter clones. The cell-cell variability can be readily measured by quantifying the luciferase signal from individual cells.

We fully agree with the reviewer that the cell-to-cell variability of GBS activity would be interesting to study. We have started to perform experiments in this direction and hope to present these findings at some point in a follow-up study.

7. For the data shown in Figure 7, how were the GR mutant variants introduced into what cells? I couldn't find the description about how these mutants were expressed. If by transient transfection, the results are affected by the highly variable expression level. In addition, depending on the cell context, the endogenous GR can interact with the exogenous mutant GR. What controls were performed to address these issues?

GR wild type and mutant variants were transiently transfected into U2OS cells, which lack endogenous GR (no reporter activity is observed in the absence of co-transfected GR expression construct). All GR variants are expressed using the same expression vector and thus, in principle, should result in a similar

variable expression level for individual cells within the population of cells examined. Notably, for other response elements, the dimerization mutant is less active than wild type indicating that the effects of mutating the dimerization interface are context-specific. To assure reproducible results, we preformed biological triplicates of each experiment.

Minor comments:

1. Page 3 line 82-83, "... post-translational modifications of DNA..." needs to be revised.

Thank you for picking up this error, we have revised this sentence.

2. In Figure 2A, I could not find the number of genes in the control group of weak responders in the legend or the main text. In addition, by the control group do the authors mean constitutively expressed genes or induced genes with very low fold change? The intended meaning is not clear.

To address this issue, we have revised a sentence in the main text describing this analysis as follows: "Therefore, we first grouped genes regulated by GR in U2OS cells¹⁵, a human osteosarcoma cell line, into strong responders (top 20% with greatest fold induction upon dexamethasone treatment, 290 genes) and a control group of weak responders (genes with significant changes in expression, log2 fold change <0.72, 688 genes) (Fig. 2A)."

3. Another incidence of overstating a minority view in the field is seen on page 20 in the paragraph starting in line 509. The authors cite a few selected studies to make general statements that are not widely believed: "...the number of occupied binding sites...correlates with the magnitude of activation."; "...genes with multiple TF binding sites tend to have a greater degree of cell-to-cell variability..." I suggest that these statements must be rephrased to indicate that there is currently a limited amount of data supporting these ideas.

We have rephrased this section as follows to avoid overstating based on a limited number of studies: "... The activity of TFs towards individual target genes can be modulated by a variety of mechanisms other than the sequence identity of the binding site. For example, a recent study showed that the number of occupied NF- κ B binding sites associated with a gene correlates with the magnitude of activation³⁹. However, in addition to being expressed³⁹ at higher levels, genes with multiple TF binding sites might display a greater degree of cell-to-cell variability (transcriptional noise) of gene expression⁴⁰. Therefore, we speculate that it could be beneficial for some GR target genes to be under control of a single, highly-active GBS with little transcriptional noise rather than multiple GBSs which induce greater noise."

Reviewer #3

Major comments:

(1) Equal DNA binding: the final mutants shown in Fig. 7B being crucial for the story, the authors should verify that DNA binding affinity/occupancy of the GR mutants is not affected by the flanking nucleotides.

We are grateful to the reviewer for suggesting these experiments and agree that they are important. We have now performed EMSA experiments using the GRdim DBD for the Cgt-GBS variant with different

flanking nucleotides and added the results to the main text (p. 15): “Comparison of the binding affinity for A/T and G/C flanked Cgt showed that GR’s affinity was comparable for both sequences for both wild type (Fig . 3B) and also for A477T DBD (A/T: 3.1 ± 0.4 μ M; G/C: 3.5 ± 1.1 μ M) although the affinity was lower for the mutant.”

(2) Are there natural polymorphisms? Can polymorphisms in flanking nucleotides be identified and linked (speculatively or not) with altered GR responses?

In principle, this is possible and likely polymorphisms exist that map to the flanking nucleotides of GBSs. However, in practice finding such candidate variants that alter GR responses requires a very large number of datasets, which we do not have at the moment (the experiments should measure the transcriptional responses to GR signaling in cells with different genetic backgrounds). The large amount of data is needed to have enough statistical power, given that the expected effect size might be small.

Minor comments:

(3) Abstract: the expression "binds short DNA fragments" does not seem appropriate for the binding mode of a TF within the context of genomic DNA.

We changed “short DNA fragments” to “genomic response elements” to rectify this issue.

(4) Fig. 2: it should be clearly indicated in the legend that these are transient transfection assays.

To make this clear, we added this information to the legend of Figure 2.

(5) Fig. 2C: this panel is too cryptic; it is difficult to read (notably also because it is not obvious what the wild-type sequences of these sites are).

To address this, we revised figure 2C, which now shows the complete sequence for each of the data points presented.

(6) Fig. 4: it should be clearer from the legend (title and contents) that the DNA shape is calculated/predicted.

We have adjusted the legend of figure 4, which now states that the DNA shape features are predictions in both the title and the content of both figure panels. Further, we added the following sentence and reference to the main text (page 10):“The DNA shape features were predicted using a high-throughput method that has been extensively validated based on experimental data (Zhou et al. NAR 2013)”..... This method was extensively validated based on all available X-ray crystallography, NMR spectroscopy, and hydroxyl radical cleavage data.

Reviewer #1 (Remarks to the Author):

In the original review I raised some issues that were mainly concerned with the rather speculative structural interpretation of the reported effect of flanking base-pairs on GR-dependent gene activation. These points have now been addressed by the authors and the structural interpretation has been largely toned down. Since the basic finding of the flanking base-bair effect is of interest, I recommend acceptance of the manuscript for publication in this form.

Reviewer #2 (Remarks to the Author):

Most of the points I raised in the last review have been addressed satisfactorily. I only have a few remaining comments:

A. The authors provided the information requested in my last major comment, #7. However, Figure 7 still does not specify that the data are from transiently transfected U2OS cells. It should be included in the legend, not just in the author response to reviewer comments.

B. Regarding the lack of endogenous GR in U2OS, have the authors confirmed by western blot that no GR is present in U2OS indeed? I think some variants of U2OS can express GR, and it would be good to check.

Reviewer #3 (Remarks to the Author):

The authors have very adequately revised their manuscript and thoroughly answered all relevant comments (mine and those of the other reviewers).

Response to reviewer's comments:

Reviewer #2

Comment A: The authors provided the information requested in my last major comment, #7. However, Figure 7 still does not specify that the data are from transiently transfected U2OS cells. It should be included in the legend, not just in the author response to reviewer comments.

We apologize for not including this information in our first revision. We revised the legend of figure 7, which now explicitly states that the data shown is derived from experiments with transiently transfected reporters.

Comment B: Regarding the lack of endogenous GR in U2OS, have the authors confirmed by western blot that no GR is present in U2OS indeed? I think some variants of U2OS can express GR, and it would be good to check.

What we have to done in the past to address this is that we have tested whether transiently transfected reporters are regulated in response to dexamethasone in the parental U2OS cells we use in our lab (without transfected GR expression construct). Similarly, we have assayed whether endogenous GR target genes are responsive to dexamethasone treatment in the parental U2OS cells. These experiments show no response to dexamethasone treatment for either reporters or for endogenous target genes, and thus that the parental U2OS line we use effectively behaves like a GR knock out.